# Evaluating Object Hallucination in Large Vision-Language Models

**Yifan Li**[1,3][*], **Yifan Du**[1,3][*], **Kun Zhou**[2][*], **Jinpeng Wang** [4],
**Wayne Xin Zhao**[2,3][†] and **Ji-Rong Wen**[1, 2,3]

[1]Gaoling School of Artificial Intelligence, Renmin University of China
[2]School of Information, Renmin University of China.
[3]Beijing Key Laboratory of Big Data Management and Analysis Methods
[4]Meituan Group
{liyifan0925, yifandu1999, batmanfly}@gmail.com,
francis_kun_zhou@163.com, wangjinpeng04@meituan.com, jrwen@ruc.edu.cn

## Abstract

Inspired by the superior language abilities of large language models (LLM), large vision-language models (LVLM) have been recently proposed by integrating powerful LLMs for improving the performance on complex multi-modal tasks. Despite the promising progress on LVLMs, we find that they suffer from object hallucinations, *i.e.,* they tend to generate objects inconsistent with the target images in the descriptions. To investigate it, this work presents the first systematic study on object hallucination of LVLMs. We conduct the evaluation experiments on several representative LVLMs, and show that they mostly suffer from severe object hallucination issues. We further discuss that the visual instructions may influence the hallucination, and find that: objects that frequently appear in the visual instructions or co-occur with the image objects are obviously prone to be hallucinated by LVLMs. Besides, we further design a polling-based query method called *POPE* for better evaluation of object hallucination. Experiment results show that our POPE can evaluate object hallucination in a more stable and flexible way.

## 1 Introduction

Large language models (LLMs) (Zhao et al., 2023) have shown remarkable abilities to solve various complex tasks by following human instructions in a zero-shot manner. The success of LLMs drives the researchers to devise more powerful multi-modal models based on the superior capacity of LLMs, to enhance the understanding of visual semantics (Alayrac et al., 2022; Li et al., 2023b). As an exemplified work, GPT-4 (OpenAI, 2023) has exhibited the exciting performance of LLMs on multimodal tasks and scenarios.

Following this line of research, a surge of studies (Zhu et al., 2023; Gao et al., 2023; Li et al.,

---

[*]Equal contribution.
[†]Corresponding author.

2023a) have been proposed to enhance the vision-language pre-trained model (VLPM) (Gan et al., 2022) by incorporating powerful LLMs (Touvron et al., 2023; Chiang et al., 2023), which are called *large vision-language model (LVLM)*. Typically, existing work reuses the visual encoder in VLPMs to handle image data, while replacing the original language encoder with LLMs. After vision-language pre-training (Alayrac et al., 2022; Li et al., 2022b) and visual instruction tuning (Liu et al., 2023), LVLMs can fulfill complex tasks according to human instructions, demonstrating strong capacities in solving various vision-language tasks, *e.g.,* image captioning (Ordonez et al., 2011; Hodosh et al., 2015; Sharma et al., 2018; Agrawal et al., 2019) and visual question answering (Antol et al., 2015a; Zhang et al., 2016; Goyal et al., 2017).

Despite the success of LVLMs, previous work has revealed that their main components, *i.e.,* LLMs and VLPMs, both suffer from hallucination. Especially, LLMs tend to hallucinate unintended text (Huang et al., 2021; Bang et al., 2023), and VLPMs might generate nonexistent objects in the image (Biten et al., 2022) (termed as *object hallucination*). It is generally believed that the hallucination would degrade the model performance and greatly harm the user experiences in real-world applications (MacLeod et al., 2017; Ji et al., 2022). Therefore, it is natural to ask the question: *does hallucination still exist in LVLMs*? In this paper, we systematically evaluate the issue of *object hallucination* in existing LVLMs, which refers to generating contents that are inconsistent with ground-truth objects in the given image.

To conduct our study, we first use the CHAIR (*Caption Hallucination Assessment with Image Relevance*) metric (Rohrbach et al., 2018), and examine the hallucination degree of several representative LVLMs on the MSCOCO dataset. Our preliminary experiments (Table 1) show that most of LVLMs severely suffer from object hallucina-

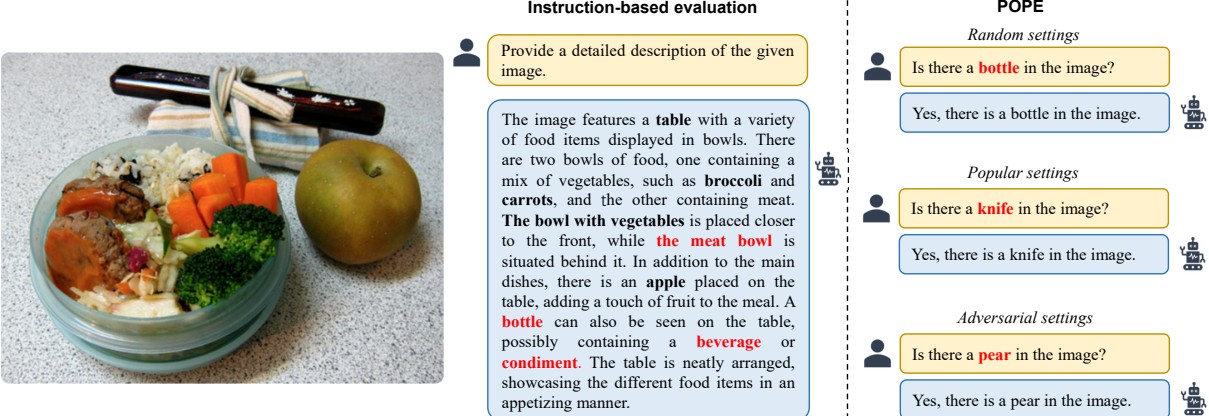

Figure 1: Cases of object hallucination in LVLMs. **Bold** objects are ground-truth objects in the annotations and red objects are hallucinated objects by LVLMs. The left case is from the traditional instruction-based evaluation method, and the right cases are from three variants of POPE.

tion, and are even more prone to hallucinate than small vision-language models. Besides, we find that the existing object hallucination evaluation method may not be best suited for LVLMs and further propose a *Polling-based Object Probing Evaluation (POPE)* method. The basic idea is to convert the evaluation of hallucination into a binary classification task by prompting LVLMs with simple *Yes*-or-*No* short questions about the probing objects (*e.g.,* Is there *a car* in the image?). We show that such a method is more stable and flexible. Besides, by using different object sampling strategies, we validate that existing LVLMs are prone to hallucinate objects which frequently appear or co-occur in the visual instruction dataset.

Our main contributions are as follows: (1) We conduct an empirical study on object hallucination for several representative LVLMs and find that they are highly affected by object hallucination. (2) We discuss the potential reasons behind this promblem, *e.g.,* LVLMs tend to generate frequently appearing or co-occurring objects in the instruction corpora. (3) We propose an object hallucination evaluation approach called POPE, which is more stable and can be easily extended to unannotated datasets.

## 2 Background

### 2.1 Large Vision-Language Model

Since LLMs have been shown to be general task solvers in a zero-shot/few-shot manner, a number of studies are devoted to improving VLPM by integrating powerful LLMs for more accurate language understanding and generation (Zhu et al., 2023; Liu et al., 2023; Dai et al., 2023a). In this paper, we re-

fer to the enhanced VLPMs with the integration of LLMs as *Large Vision-Language Models (LVLM)*.

Generally speaking, an LVLM consists of a vision encoder, a language encoder (*i.e.,* an LLM), and a cross-modal alignment network. The training of LVLMs is generally composed of three major steps. First, a vision encoder and a language encoder are pre-trained on large-scale unimodal data (*i.e.,* image and text data, respectively). Second, these two encoders are aligned through image-text alignment pre-training, which enables the LLM to generate a meaningful caption for a given image. Third, the aligned model is further fine-tuned on image-text instructions, so that it can generate satisfactory answers *w.r.t.* to a natural language question regarding a specific image. Note that in the second and third steps, we can optionally fine-tune different components instead of performing full-parameter fine-tuning.

Once the visual encoder and the LLM are well aligned, the derived LVLM can demonstrate a superior visual understanding ability. It can not only grasp the visual semantics of objects in the image, but also deeply understand the linguistic semantics for these objects by leveraging the parametric knowledge in the LLM. Further, the LVLM can perform complex reasoning over the related concepts about these objects, thus achieving an improved performance on a variety of multimodal tasks, *e.g.,* visual question answering (VQA).

### 2.2 Object Hallucination

Although LVLMs are powerful in solving vision-language tasks, they also suffer from the issue of *object hallucination* as VLPMs. In the literature

of computer vision field (Rohrbach et al., 2018; Biten et al., 2022), object hallucination refers that the model generating descriptions or captions that contain objects which are inconsistent with or even absent from the target image. In general, object hallucination can be defined at different semantic levels. The most straightforward way is to define it over the object level, while more fine-grained definitions might be concerned with the attributes or characteristics of objects. In this work, we focus on coarse-grained object hallucinations in the model-generated captions and leave fine-grained object hallucinations such as the number, attributes, and positions of the object for future work. We present an example of object hallucination in Figure 1, where the hallucinated object "meat bowl", "bottle", "beverage", "condiment" are generated by the underlying LVLMs.

The hallucination phenomenon hinders the safe use of LVLMs in real-world deployment, as it may result in unexpected consequences caused by these hallucinated objects (MacLeod et al., 2017). For example, due to an incorrect understanding of the external environment, an autonomous driving system would make wrong decisions when encountering unexpected events, which might lead to serious safety issues. In order to mitigate these issues, this work aims to study how object hallucination exists in LVLMs from an evaluation perspective.

## 3 Object Hallucination in LVLMs

In this section, we evaluate the object hallucination problem in popular LVLMs using an existing method. We first introduce the evaluation settings and then analyze the experimental results.

### 3.1 Evaluation Settings

Caption Hallucination Assessment with Image Relevance (CHAIR) (Rohrbach et al., 2018) is a popular metric for evaluating object hallucination in image captioning tasks. Given the ground truth objects in the image, CHAIR calculates the proportion of objects that appear in the caption but not the image. Existing work commonly adopts its two variants, *i.e.*, $\text{CHAIR}_I$ and $\text{CHAIR}_S$, which evaluate the hallucination degree at the object instance level and sentence level respectively. They can be formulated as:

$$\text{CHAIR}_I = \frac{|\{\text{hallucinated objects}\}|}{|\{\text{all mentioned objects}\}|}, \quad (1)$$

$$\text{CHAIR}_S = \frac{|\{\text{captions with hallucinated objects}\}|}{|\{\text{all captions}\}|}. \quad (2)$$

| I | Model | $\text{CHAIR}_I$ | $\text{CHAIR}_S$ | Len |
|---|---|---|---|---|
| - | $\text{OSCAR}_{Base}$ | 7.1 | 13.0 | - |
| | $\text{VinVL}_{Large}$ | 5.5 | 10.5 | - |
| | $\text{OFA}_{Large}$ | **4.7** | 8.9 | - |
| | $\text{BLIP}_{Large}$ | **4.7** | **8.8** | - |
| $I_1$ | mPLUG-Owl | 14.8 | 25.4 | 35.8 |
| | LLaVA | 10.5 | 32.7 | 64.3 |
| | MultiModal-GPT | 11.1 | 15.0 | 11.6 |
| | MiniGPT-4 | 6.7 | 9.5 | 24.7 |
| | InstructBLIP | **2.6** | **3.7** | 8.5 |
| $I_2$ | mPLUG-Owl | 30.2 | 76.8 | 98.5 |
| | LLaVA | 18.8 | 62.7 | 90.7 |
| | MultiModal-GPT | 18.2 | 36.2 | 45.7 |
| | MiniGPT-4 | 9.2 | 31.5 | 116.2 |
| | InstructBLIP | **2.5** | **3.4** | 7.5 |

Table 1: Results of CHAIR on VLPMs and LVLMs. $I_1$ denotes "*Generate a short caption of the image*" and $I_2$ denotes "*Provide a brief description of the given image*". Len refers to the average length of generated captions. The results of VLPMs (OSCAR, VinVL, BLIP, and OFA) are collected from Dai et al. (2023b). The best results in each block are denoted in bold.

We select five recently released LVLMs, *i.e.*, mPLUG-Owl (Ye et al., 2023), LLaVA (Liu et al., 2023), Multimodal-GPT (Gong et al., 2023), MiniGPT-4 (Zhu et al., 2023) and InstructBLIP (Dai et al., 2023a) and prompt them with following instructions to generate captions about images in MSCOCO (Lin et al., 2014):

- $I_1$: *Generate a short caption of the image.*
- $I_2$: *Provide a brief description of the given image.*

Then, we calculate CHAIR on these captions. We leave more details about the introduction to the dataset and evaluated models in Appendix A.

### 3.2 Evaluation Results

**Severity of Hallucinations.** As the evaluation results illustrated in Table 1, most instruction-tuned LVLMs suffer from the object hallucination problem, even more serious than small models, *e.g.*, LLaVA (32.7) *v.s.* $\text{OSCAR}_{base}$ (13.0) on $\text{CHAIR}_S$ using *Instruction 1*. It indicates that object hallucination is an important problem for LVLMs and deserves to be concerned about. As a comparison, InstructBLIP hallucinates less than other LVLMs. A possible reason is that its visual instructions are collected from a wide variety of publicly available datasets, which are relatively short. In contrast, other LVLMs mostly employ the visual instructions generated by unimodal LLMs (Liu et al., 2023). Such synthetic visual instructions are gener-

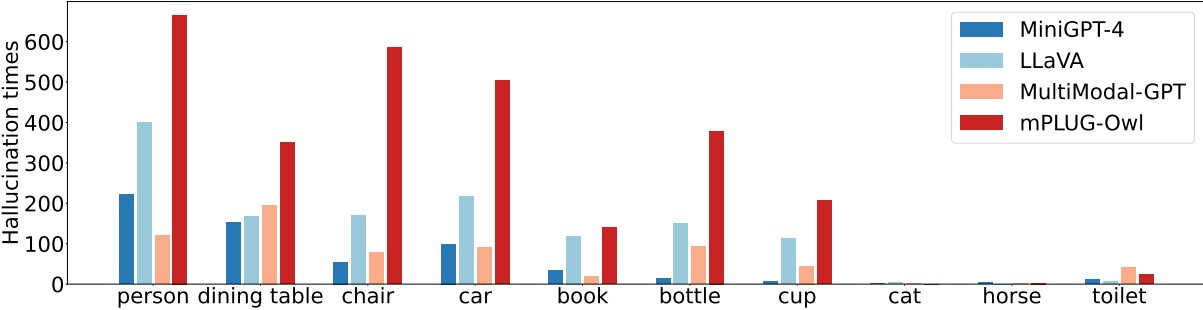

(a) Hallucination times of top ten frequently appearing objects, whose frequencies decrease from right to left.

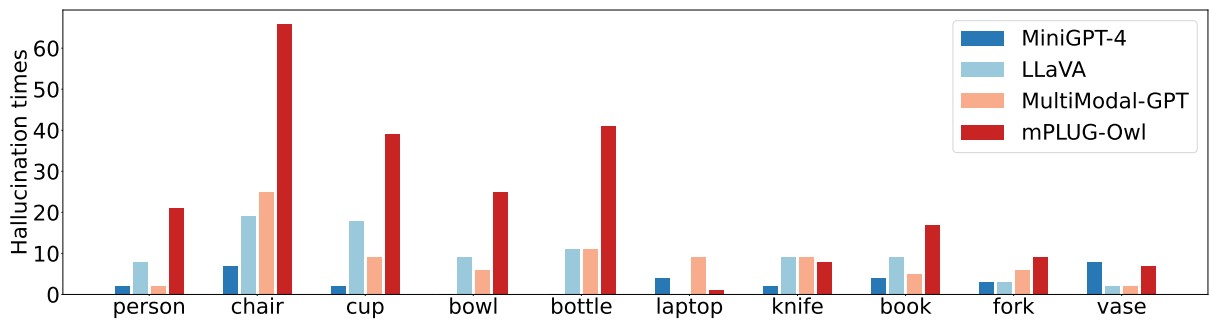

(b) Hallucination times of top ten objects co-occurring with "dining table", whose frequencies decrease from right to left.

Figure 2: Hallucination times of frequently appearing/co-occurring objects in MSCOCO.

ally longer and more informative, but may involve unexpected descriptive information (hallucination inherent from LLMs) that is inconsistent with the image, which could mislead LVLMs.

**Disadvantages of CHAIR.** As Table 1 shows, the evaluation results can be affected by other factors, *e.g.,* instruction designs and the length of captions. Specifically, although the adopted two instructions have similar semantic meanings, LVLMs prompted by *Instruction 2* can even result in doubled values of CHAIR metrics compared with those prompted by *Instruction 1*, and the performance order of some LVLMs also changes (*e.g.,* $CHAIR_I$ values of LLaVA and MultiModal-GPT). It indicates the instability of the CHAIR metric when different instructions are employed. Besides, as CHAIR requires to examine whether the mentioned objects are hallucinated in the generated caption, it needs complex human-crafted parsing rules to perform exact matching, which has not been adapted to the special generation styles of LVLMs and may lead to misclassification errors.

Thus, it is necessary to consider a more suitable method that can stably and conveniently evaluate the object hallucination problem in LVLMs.

## 4 Influence of Instruction Data on Object Hallucination

Considering their impressive performance on complex vision-language tasks (Chen et al., 2023; Bai et al., 2023; Li et al., 2023a), it is counter-intuitive that the hallucination problem of LVLMs is so severe. Since smaller VLPMs suffer less from object hallucination, it is possible that the visual instruction-tuning process of LVLMs exacerbates object hallucination. In this section, we investigate the influence of the visual instruction data. We first make two basic hypotheses in Section 4.1 and then conduct qualitative and quantitative analysis to verify them in Section 4.2 and Section 4.3.

### 4.1 Hypotheses

As the visual instruction datasets of these LVLMs are mostly constructed based on MSCOCO (Lin et al., 2014), they generally share a similar unbalanced object distribution where top frequent objects occupy a major part of the dataset. After being fine-tuned on them, LVLMs may also be prone to generate (or hallucinate) frequently appearing objects in MSCOCO. Additionally, the presence of frequently co-occurring object groups (*e.g.,* laptop, mouse and keyboard) may also contribute to

| Model | HR$_A$ | | | HR$_C$(dining table) | | |
|---|---|---|---|---|---|---|
| | @10 | @20 | @30 | @10 | @20 | @30 |
| mPLUG-Owl | 0.5455 | 0.6591 | 0.7533 | 0.6608 | 0.7926 | 0.8253 |
| LLaVA | 0.4620 | 0.5911 | 0.6796 | 0.5628 | 0.7329 | 0.8595 |
| MultiModal-GPT | 0.4152 | 0.5399 | 0.6743 | 0.5742 | 0.7849 | 0.8961 |
| MiniGPT-4 | 0.4610 | 0.5758 | 0.7207 | 0.5600 | 0.6980 | 0.9145 |

Table 2: Results on MSCOCO that quantify the correlations between the appearing/co-occurring frequency of objects and the hallucination times of LVLMs.

object hallucination. LVLMs can be elicited by the existing objects in the image to hallucinate other objects that frequently co-occur with them. Therefore, we hypothesize that (1) LVLMs are prone to hallucinate frequently appearing objects in the visual instruction datasets; (2) LVLMs are prone to hallucinate objects that frequently co-occur with ground-truth objects in the image. We conduct qualitative and quantitative analyses in the following parts to verify them.

### 4.2 Qualitative Analysis

We first qualitatively analyze the correlation between the appearance frequency and hallucination. For the first hypothesis, we plot a bar chart between the top ten frequently appearing objects in MSCOCO and their hallucination times in the validation set of MSCOCO; for the second hypothesis, we select the top ten frequently co-occurring objects with "dining table" and also plot a bar chart to show their hallucination times across images that really contain "dining table". We show the results of MiniGPT-4, LLaVA, MultiModal-GPT and mPLUG-Owl in Figure 2. Obviously, with the decreasing of the occurrence frequency of objects (from right to left), there is a notable decrease in the hallucination times for all four LVLMs. It reveals that the frequently appearing and co-occurring objects in the visual instruction dataset are indeed more likely to be hallucinated by LVLMs. To better support our results, we also list the full statistics of all 80 COCO objects in Appendix B.

### 4.3 Quantitative Analysis

To further consolidate the above findings, we employ the top-$k$ hit ratio (HR@$k$) to measure the consistency between the appearance frequency and hallucination times of objects, which is defined as:

$$\text{HR}_A@k = \frac{1}{n} \sum_{i=1}^{n} \frac{\text{Hit@}k(i)}{\text{Hallucinated}(i)}, \quad (3)$$

$$\text{HR}_C@k(o) = \frac{1}{m} \sum_{i=1}^{m} \frac{\text{Hit@}k(i,o)}{\text{Hallucinated}(i)}, \quad (4)$$

where HR$_A$ and HR$_C$ quantify the correlations between hallucination times and appearing and co-occurring frequency respectively. $n$ is the total number of images, Hallucinated($i$) denotes the number of hallucinated objects in the $i$-th example, Hit@$k(i)$ denotes the number of top-$k$ frequently appearing MSCOCO objects in Hallucinated($i$), and Hit@$k(i,o)$ denotes the number of top-$k$ frequently co-occurring objects with the probing object $o$ in Hallucinated($i$). Therefore, HR@$k$ can reflect the proportion of top-$k$ frequently appearing or co-occurring objects in all hallucinated objects.

We present the HR$_A$ and HR$_C$(dining table) of top 30 objects in Table 2 and leave HR$_C$@(chair) and HR$_C$@(car) in Appendix C. The HR$_A$@10 and HR$_C$@10(dining table) of all LVLMs are near 0.5 and 0.6, respectively. It indicates that, on average, approximately half of the hallucinated objects in each image belong to the top 10 frequently appearing COCO objects, while more than half are among the top 10 frequently co-occurring objects with the objects already present in the image. When we broaden our observation to the top 30 objects, this proportion continues to increase. These findings further verify that LVLMs mostly hallucinate common objects in the visual instruction data and inspire us to design three sampling strategies in our evaluation pipeline.

## 5 POPE

In this section, we devise *Polling-based Object Probing Evaluation (POPE)*, a simple yet effective approach for evaluating hallucination in LVLMs. We first provide an overview of POPE, and then evaluate the representative LVLMs with POPE. Finally, we discuss the stability and scalability of our method, and also analyze the impact of hallucina-

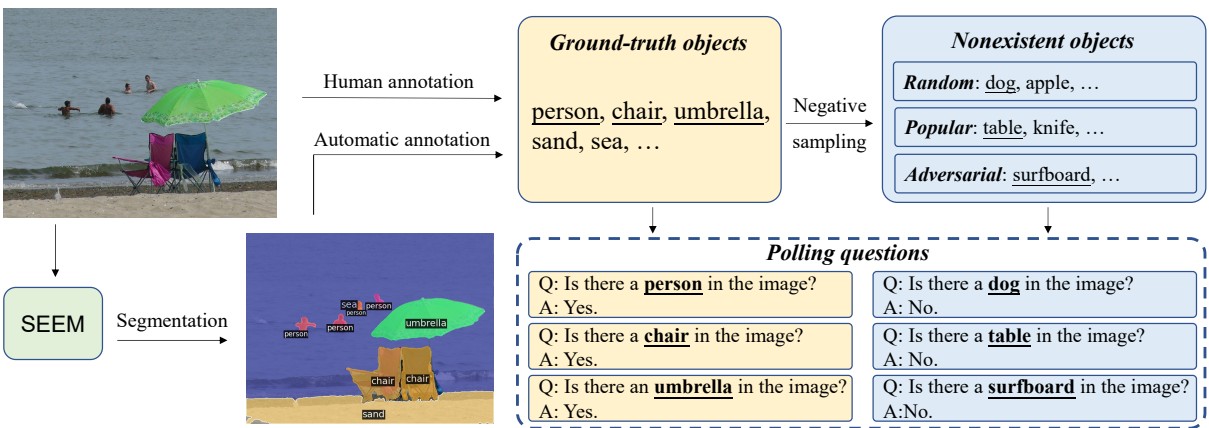

Figure 3: Overview of the POPE pipeline. Given an input image, POPE first extracts ground-truth objects in the image either from human annotations or with the help of automatic segmentation tools like SEEM. Then, POPE conducts negative sampling for nonexistent objects in the image under *Random/Popular/Adversarial* settings. Finally, the ground-truth objects and nonexistent objects are formulated into question templates to poll LVLMs.

tion on VQA task.

## 5.1 Overview of POPE

In the empirical results of Section 3, we have revealed the severity of the object hallucination problem in LVLMs and highlighted the limitations of the existing evaluation method, *e.g.,* sensitive to instructions and biased to short captions. Besides, existing methods mostly rely on parsing the generated captions to extract the predicted objects, which usually require human-crafted complex rules and are still inevitable to omit or misclassify objects.

Therefore, we consider devising a more suitable method for the stable, fair and flexible object hallucination evaluation of LVLMs, namely polling-based object probing evaluation (POPE). Specifically, POPE formulates the evaluation of object hallucination as a binary classification task that prompts LVLMs to output "Yes" or "No", *e.g., "Is there a* chair *in the image?"*. In this way, by sampling objects that LVLMs are prone to hallucinate, we can construct a set of hard questions to poll LVLMs. As standard answers to these questions are just "Yes" or "No", we can easily identify them without complex parsing rules, and avoid the influence of instruction designs and caption length, thus guaranteeing stability, fairness and flexibility.

**Definition.** Given an image caption dataset, POPE focuses on constructing a set of triples, each of which consists of an image, multiple questions and their answers ("Yes" or "No"). The formulated definition of a triple can be described as:

$$\langle x, \{q(o_i), a_i\}_{i=1}^{l}\rangle, \quad (5)$$

where $x$ denotes the image, $q(o_i)$ is the question probing $o_i$ based on a template *"Is there a/an* <object> *in the image?"*, $o_i$ is the $i$-th object to be probed, $a_i$ is the answer to the question ("Yes" or "No") and $l$ denotes the number of polling questions per image. $o_i$ can be obtained either from annotations or the results of automatic segmentation tools like SEEM (Zou et al., 2023). We set the ratio between ground-truth and nonexistent objects as 1:1 for label balance. After constructing the evaluation triples, we can directly poll LVLMs with them and collect the predicted answers.

**Pipeline.** The whole POPE pipeline is presented in Figure 3. After obtaining objects in the image, we can start to building polling questions. Questions whose answers are "Yes" can be directly built using ground-truth objects, while questions with the answer "No" can be built by sampling from negative objects. Therefore, by devising different sampling strategies, we can validate whether LVLMs are prone to hallucinate specific objects, *e.g.,* frequently appearing or co-occurring objects discussed in Section 4. Thus, we devise the following three sampling strategies:

● **Random Sampling**: we randomly sample the objects that do not exist in the image.

● **Popular Sampling**: we select the top-$k$ most frequent objects in the whole image dastaset that do not exist in the current image, where $k = \frac{l}{2}$.

● **Adversarial Sampling**: we first rank all objects according to their co-occurring frequencies with the ground-truth objects, and then select the top-$k$ frequent ones that do not exist in the image.

| Dataset | POPE | Model | Accuracy | Precision | Recall | F1 Score | Yes (%) |
|---------|------|-------|----------|-----------|--------|----------|---------|
| MSCOCO | *Random* | mPLUG-Owl | 53.30 | 51.71 | 99.53 | 68.06 | 96.23 |
| | | LLaVA | 54.43 | 52.32 | 99.80 | 68.65 | 95.37 |
| | | MultiModal-GPT | 50.03 | 50.02 | **100.00** | 66.68 | 99.97 |
| | | MiniGPT-4 | 77.83 | 75.38 | 82.67 | 78.86 | 54.83 |
| | | InstructBLIP | **88.73** | **85.08** | 93.93 | **89.29** | 55.20 |
| | *Popular* | mPLUG-Owl | 50.63 | 50.32 | 99.27 | 66.79 | 98.63 |
| | | LLaVA | 52.43 | 51.25 | 99.80 | 67.72 | 97.37 |
| | | MultiModal-GPT | 50.00 | 50.00 | **100.00** | 66.67 | 100.00 |
| | | MiniGPT-4 | 68.30 | 64.27 | 82.40 | 72.21 | 64.10 |
| | | InstructBLIP | **81.37** | **75.07** | 93.93 | **83.45** | 62.57 |
| | *Adversarial* | mPLUG-Owl | 50.67 | 50.34 | 99.33 | 66.82 | 98.67 |
| | | LLaVA | 50.77 | 50.39 | 99.87 | 66.98 | 99.10 |
| | | MultiModal-GPT | 50.00 | 50.00 | **100.00** | 66.67 | 100.00 |
| | | MiniGPT-4 | 66.60 | 62.45 | 83.27 | 71.37 | 66.67 |
| | | InstructBLIP | **74.37** | **67.67** | 93.33 | **78.45** | 68.97 |

Table 3: Results of LVLMs under three evaluation settings of POPE on the validation set of MSCOCO. Yes denotes the proportion of answering "Yes" to the given question. The best results in each block are denoted in bold.

Under the above three settings, we can build the evaluation questions of different difficulty levels. We evaluate previously mentioned LVLMs on them with the following metrics.

**Metrics.** We adopt Accuracy, Precision, Recall and F1 score as the evaluation metrics. Accuracy reflects the proportion of correctly answered questions. Precision and Recall reflect the ratios of correctly answering questions whose answers are "Yes" or "No", respectively. F1 score combines the results of Precision and Recall and we select it as the major metric for evaluation. Besides, we also report the ratio that LVLMs answer "Yes" as a reference to analyze the model behaviors.

## 5.2 Evaluation on MSCOCO

We evaluate all the LVLMs with POPE built on the validation set of MSCOCO (Lin et al., 2014). We randomly select 500 images with more than 3 ground-truth objects in the annotations and construct 6 questions for each image (*i.e.*, $l = 6$).

The results are presented in Table 3, where we can obtain a similar conclusion as in Table 1 that InstructBLIP performs the best, while LLaVA, MultiModal-GPT and mPLUG-Owl suffer more severe hallucination problem, whose F1 Score are below 70. It indicates that POPE can well estimate the degree of the hallucination problem in LVLMs. Besides, we find that LLaVA, MultiModal-GPT and

mPLUG-Owl are extremely prone to answer "Yes" (near 99%). It reveals that these three LVLMs are *over confident*, leading to lower accuracy on questions with the answer "No". Furthermore, the performance of LVLMs consistently decreases, from random settings, to popular and adversarial. It is consistent with our findings in Section 4, as LVLMs are prone to hallucinate the frequently appearing and co-occurring objects.

## 5.3 Advantages of POPE

As previously stated, the current approach for evaluating object hallucination in LVLMs like CHAIR is instruction-based, which is hindered by LVLMs' sensitivity to prompts and requires object annotations and manually designed rules for evaluation. In contrast, POPE is more stable to prompt forms and can be easily extended to unannotated datasets. Its probing result is also highly consistent with model's caption.

**Stability.** Regardless of the variations in prompt templates, POPE requires LVLMs to answer simple closed-ended questions, which is less likely to introduce ambiguity compared to instruction-based methods. Such characteristic contributes to its stability. To validate it, we evaluate LLaVA using both POPE and CHAIR$_I$ with four different prompts for each. The evaluation results are presented in Table 4. It can be observed that the standard deviation

| POPE | | CHAIR | |
|---|---|---|---|
| **Prompt** | F1 Score | **Prompt** | CHAIR$_I$ |
| Is there a <object> in the image? | 68.65 | Generate a short caption of the image. | 10.50 |
| Does the image contain a <object>? | 66.83 | Provide a brief description of the image. | 18.80 |
| Have you noticed a <object> in the image? | 66.67 | Generate a concise description for the image. | 14.60 |
| Can you see a <object> in the image? | 67.58 | Create a short textual summary for the image. | 11.60 |
| Avg±Std. | 67.43±0.78 | | 13.88±3.22 |

Table 4: Evaluation results of LLaVA on POPE and CHAIR with different prompt templates.

| Dataset | POPE | Model | Accuracy | Precision | Recall | F1 Score | F1 Score (Truth) | Yes (%) |
|---|---|---|---|---|---|---|---|---|
| MSCOCO | *Random* | LLaVA | 50.47 | 50.24 | **99.67** | 66.80 | 68.65 | 99.20 |
| | | MiniGPT-4 | 73.77 | 79.25 | 64.40 | 71.06 | 78.86 | 40.63 |
| | | InstructBLIP | **86.60** | **80.74** | 96.13 | **89.29** | 89.27 | 59.53 |
| | *Popular* | LLaVA | 50.00 | 50.00 | **99.27** | 66.50 | 67.72 | 99.27 |
| | | MiniGPT-4 | 67.80 | **68.80** | 65.13 | 66.92 | 72.21 | 47.33 |
| | | InstructBLIP | **71.27** | 64.20 | 96.13 | **76.99** | 83.45 | 74.87 |
| | *Adversarial* | LLaVA | 49.77 | 49.88 | **99.20** | 66.38 | 66.98 | 99.43 |
| | | MiniGPT-4 | 61.93 | **61.46** | 64.00 | 62.70 | 71.37 | 52.07 |
| | | InstructBLIP | **62.53** | 57.50 | 96.13 | **71.96** | 78.45 | 83.60 |

Table 5: SEEM-based POPE results of LVLM on MSCOCO. F1 Score (Truth) are the results of POPE using ground-truth annotations, which are copied from Table 3. The best results in each block are denoted in bold.

of the F1 score is significantly lower than CHAIR$_I$, which confirms that POPE exhibits higher stability when faced with different prompts.

**Scalability.** As mentioned before, with the assistance of automatic segmentation tools, POPE can be easily extended to datasets without annotations. To validate it, we adopt SEEM (Zou et al., 2023) to annotate images from three datasets (*i.e.,* MSCOCO, A-OKVQA (Schwenk et al., 2022) and GQA (Hudson and Manning, 2019)) and build POPE based on the segmentation results. We evaluate InstructBLIP, MiniGPT-4 and LLaVA on them and report the results in Table 5 and Table 11 (presented in Appendix D). In Table 5, the performances of all LVLMs mostly follow the same trend as annotation-based POPE in Table 3, *i.e., Random > Popular > Adversarial*, and *InstructBLIP > MiniGPT-4 > LLaVA*. Such consistency indicates the reliability of the SEEM-based POPE. Whereas, we also notice the performance gap between the two settings, *e.g.,* F1 Score 71.37 *v.s.* 62.70 for MiniGPT-4 under the *Adversarial* setting. This phenomenon can be attributed to the finer granularity of the segmentation results generated by SEEM, which makes the POPE more challenging. In summary, when combined with automated segmentation tools, POPE can be easily extended to unannotated datasets and conduct effective evalua-

tions on them.

**Consistency.** A potential concern for POPE is whether the Yes/No responses of LVLMs genuinely reflect their perception of objects. To validate this, we measure the consistency between the POPE responses and captions generated by LVLMs. Specifically, we examine if objects that receive "No" responses seldom appear in the captions, and if objects frequently mentioned in captions usually receive "Yes" answers. We collect data from InstructBLIP and MiniGPT-4, given their relatively balanced yes/no distributions. Our findings reveal that out of the 1303 and 1445 objects that are given "No" responses by InstructBLIP and MiniGPT-4, merely 0 and 5 of those objects were referenced in captions. Moreover, out of the 664 and 1034 objects mentioned in the captions by these models, 664 and 961 respectively received a "Yes" verdict. Such results underscore a robust correlation between objects' presence in captions and Yes/No responses in POPE questions about them, validating the reliability of the POPE assessment.

### 5.4 Impact of Hallucination on Vision Tasks

Although existing LVLMs do suffer from significant object hallucination issues, it remains an open question whether these hallucinations have a strong impact on other vision tasks. Therefore, we compare their performance on POPE with VQA and

| Dataset | Model | POPE↑ | VQA↑ |
|---------|-------|-------|------|
| A-OKVQA | InstructBLIP | **87.20** | **59.68** |
| | MiniGPT-4 | 72.47 | 38.69 |
| | LLaVA | 66.64 | 50.51 |
| GQA | InstructBLIP | **85.32** | **62.12** |
| | MiniGPT-4 | 67.13 | 42.24 |
| | LLaVA | 66.56 | 47.60 |

Table 6: Evaluation results of LVLMs on POPE and VQA. For VQA tasks, we report the VQA score on A-OKVQA and Accuracy on GQA. For POPE, we copy the result under the random setting from Table 11.

image captioning tasks. For VQA tasks, we evaluate the SEEM-based POPE and VQA scores of LVLMs on A-OKVQA and GQA datasets. Since LVLMs are prone to generate answers in an open-ended manner, we utilize ChatGPT to help parse the generated results to better evaluate the VQA performance. The details of evaluation settings are presented in Appendix E. For image captioning tasks, we evaluate the captions of 500 images in POPE with traditional metrics. The evaluation results are left in Appendix F.

The evaluation results are shown in Table 6. InstructBLIP performs the best under all settings, highlighting the importance of instruction-tuning on large visual instruction corpora. Note that since InstructBLIP has been trained on A-OKVQA, the result should be considered with caution. Furthermore, despite MiniGPT-4 achieving a higher F1 score compared to LLaVA, its performance on VQA tasks is relatively poor. A possible reason is that the instruction dataset of MiniGPT-4 only derives from image caption data, while LLaVA uses 158K visual instructions data involving complex visual questions. The results imply that the degree of hallucination may not be always consistent with the VQA performance and these two evaluation aspects are both important and should be considered in real-world applications.

## 6 Conclusion

In this work, we conducted evaluation experiments on several LVLMs and examined how they suffer from the object hallucination issue. By investigating the reasons for object hallucination, we empirically revealed that the object distributions of the visual instructions would affect the object hallucination of LVLMs. Besides, we also found that the existing hallucination evaluation methods might be affected by the input instructions and the generated

text of LVLMs, thus leading to less reliable evaluation results. To address this issue, we proposed a polling-based query method called POPE, to provide an improved evaluation approach for the object hallucination of LVLMs. Experimental results have shown that our proposed POPE can better evaluate the object hallucination issue of LVLMs.

## 7 Limitations

Despite that we have made extensive explorations, this work still has several limitations. First, we only focus on the object hallucination problem in LVLMs, while do not consider other aspects that can reflect the capacities of LVLMs. It means that the current evaluation task cannot measure the *overall performance* of LVLMs. In other words, if some model got a higher score in our evaluation setting, it does not necessarily indicate a stronger overall capacity than the one with a lower score. Second, due to the limitation of computation resources, we have to evaluate all models on a part of the validation set for each dataset. The reported results might be affected by the corresponding data distribution, though we have carefully set up the experiments. Third, our proposed POPE utilizes a matching-based method to determine whether LVLMs answer "Yes" or "No", while empirically, LVLMs may occasionally fail to provide answers explicitly containing these words, which may lead to inaccurate evaluation results. Fourth, when combined with the automatic segmentation tool, the objects would be annotated based on the label set by the tool, which may be inconsistent with the collected human annotations, leading to a divergence in evaluation results. Finally, this work has only compared a small number of LVLMs, without including some recently released or closed-source ones. We leave the evaluation of more LVLMs as our future work.

Although we have extensively discussed the hallucination issues of LVLMs, it does not indicate that we hold an negative opinion on their progress. Instead, it will be a very promising direction to develop LVLMs by leveraging the powerful LLMs. These models that were evaluated in this work have been excellent demonstrations for this direction. While, we do hope that our work can bring new ideas or insights to develop more reliable and human-aligned LVLMs.

## Acknowledgements

This work was partially supported by National Natural Science Foundation of China under Grant No. 62222215, Beijing Natural Science Foundation under Grant No. L233008 and 4222027, and Beijing Outstanding Young Scientist Program under Grant No. BJJWZYJH012019100020098. This research was also supported by Meituan.

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

## A  Details of Evaluation Settings

**Dataset.** MSCOCO (Lin et al., 2014) is a large-scale image recognition, segmentation, and captioning dataset. Here, we randomly sample 2,000 images with annotations about contained objects and human-labeled captions from its validation set as our evaluation dataset. For computing the CHAIR metric on MSCOCO, we follow the settings in Rohrbach et al. (2018) which only considers 80 objects appearing in the MSCOCO segmentation challenge.

**Models.** The evaluated LVLMs basically consist of three parts: a visual encoder, an alignment model, and a large language model. All the above models have been tuned on collected visual instruction data. A detailed comparison (*e.g.,* backbones and trainable components) of these LVLMs is shown in Table 7. We also collect the evaluation results of smaller VLPMs, *i.e.,* OSCAR (Li et al., 2020), VinVL (Zhang et al., 2021), BLIP (Li et al., 2022b) and OFA (Wang et al., 2022) from Dai et al. (2023b) as baseline results.

## B  Additional Qualitative Analysis Results

To better validate our hypotheses, We expand the analysis scope to all 80 objects in MSCOCO and present the result in this part.

For hypothesis (1), we present the cumulative proportions of the hallucination times of all 80 COCO objects in Table 8. The table demonstrates that, across all models, the top 30 objects comprise approximately 70% of all hallucinated objects. For hypothesis (2), we present the cumulative proportions of the hallucination times of all COCO objects that co-occur with dining table in Table 9. We also arrange these objects by their co-occurrence frequency. Similarly, the top 20 objects comprise about 80% of all hallucinated objects.

## C  Additional Quantitative Analysis Results

We present the $HR_C$ results of two other common objects, *i.e.,* chair and car in Table 10, which show a similar trend with Table 2.

## D  Results of SEEM-based POPE on A-OKVQA and GQA

We adopt SEEM (Zou et al., 2023) to annotate images from A-OKVQA and GQA and build POPE

| Model | VE | AN | LLM | Pre-training | | | Fine-tuning | | |
|---|---|---|---|---|---|---|---|---|---|
| | | | | VE | AN | LLM | VE | AN | LLM |
| mPLUG-Owl | ViT-L/14 | Attention | LLaMA$_{7B}$ | 🔥 | 🔥 | ❄️ | ❄️ | ❄️ | 🔥 |
| LLaVA | ViT-L/14 | Linear | LLaMA$_{13B}$ | ❄️ | 🔥 | ❄️ | ❄️ | 🔥 | 🔥 |
| MultiModal-GPT | ViT-L/14 | Attention | LLaMA$_{7B}$ | ❄️ | 🔥 | ❄️ | ❄️ | ❄️ | 🔥 |
| MiniGPT-4 | ViT-G/14 | Linear | Vicuna$_{13B}$ | ❄️ | 🔥 | ❄️ | ❄️ | 🔥 | ❄️ |
| InstructBLIP | ViT-G/14 | Q-Former | Vicuna$_{13B}$ | ❄️ | 🔥 | ❄️ | 🔥 | 🔥 | ❄️ |

Table 7: Comparison of the evaluated LVLMs. VE, AN, LLM stand for Visual Encoder, Alignment Network and Large Language Model, respectively. ❄️ denotes *frozen* and 🔥 denotes *trainable*. The fine-tuning of LLM in MultiModal-GPT and mPLUG-Owl is implemented by LoRA.

| Model | Accumulative proportions (sorted by appearance frequency) | | | | | | | |
|---|---|---|---|---|---|---|---|---|
| | Top 10 | Top 20 | Top 30 | Top 40 | Top 50 | Top 60 | Top 70 | Top 80 |
| mPLUG-Owl | 56.89 | 67.75 | 77.34 | 79.93 | 90.41 | 95.33 | 98.75 | 100.00 |
| LLaVA | 48.47 | 60.29 | 69.74 | 72.80 | 81.46 | 90.26 | 96.69 | 100.00 |
| Multimodal-GPT | 43.07 | 56.12 | 68.85 | 72.78 | 81.59 | 90.07 | 97.07 | 100.00 |
| MiniGPT-4 | 44.96 | 55.19 | 70.68 | 78.12 | 84.36 | 93.23 | 97.59 | 100.00 |

Table 8: The accumulated proportions of the hallucination times of all 80 COCO objects. We arrange all objects by their frequency of occurrence.

| Model | Accumulative proportions (sorted by co-occurrence frequency) | | | | | |
|---|---|---|---|---|---|---|
| | Top 10 | Top 20 | Top 30 | Top 40 | Top 50 | >Top 60 |
| mPLUG-Owl | 71.78 | 82.82 | 86.81 | 92.94 | 99.08 | 100.00 |
| LLaVA | 60.69 | 73.79 | 85.52 | 93.10 | 97.24 | 100.00 |
| Multimodal-GPT | 53.50 | 75.16 | 85.99 | 96.82 | 98.09 | 100.00 |
| MiniGPT-4 | 64.00 | 80.00 | 92.00 | 94.00 | 96.00 | 100.00 |

Table 9: The accumulated proportions of the hallucination times all objects that co-occur with `dining table`. We arrange all objects by their co-occurrence frequency with `dining table`.

based on segmentation results. We evaluate InstructBLIP, MiniGPT-4 and LLaVA. We also evaluate a full-data supervised-tuned smaller model, BLIP (Li et al., 2022b) to better reflect the degree of hallucination. The evaluation results are presented in Table 11.

## E ChatGPT-assisted VQA Evaluation.

We employ the VQA score (Antol et al., 2015b) to assess VQA tasks with the help of ChatGPT. Considering that LVLMs generally produce open-ended responses, we enlist ChatGPT to assess whether the model's reply aligns with potential answers. The prompt we provide to ChatGPT is as follows:

• *"You are an examiner who can judge whether a student's answer matches the correct answers. Next, I will provide you with 10 correct answers in the form of a list and a student's answer. Please judge whether the student's answer matches one of the 10 correct answers. If it matches, please output the correct answer directly (must be an element in the list, if it matches multiple correct answers, please output the most frequent occurrence in the list); if not, please output <NAN> directly. Do NOT output anything else!*
*correct answers:*
*student answer:"*

## F Results of Image Captioning

The MSCOCO captioning results of LVLMs are showcased in Table 12. Generally, their captioning performance aligns with the POPE assessments, suggesting that object hallucination influences the efficacy of LVLMs in other vision tasks.

| Model | HR$_C$(chair) | | | HR$_C$(car) | | |
|---|---|---|---|---|---|---|
| | @10 | @20 | @30 | @10 | @20 | @30 |
| mPLUG-Owl | 0.5926 | 0.7186 | 0.8201 | 0.7587 | 0.9136 | 0.9707 |
| LLaVA | 0.5206 | 0.6830 | 0.8152 | 0.6870 | 0.8886 | 0.9188 |
| MultiModal-GPT | 0.5732 | 0.7576 | 0.8811 | 0.6031 | 0.8482 | 0.8623 |
| MiniGPT-4 | 0.5701 | 0.7746 | 0.8581 | 0.6444 | 0.8278 | 0.9417 |

Table 10: The HR$_C$ result of chair and car.

| Dataset | POPE | Model | Accuracy | Precision | Recall | F1 Score | Yes (%) |
|---|---|---|---|---|---|---|---|
| A-OKVQA | *Random* | BLIP | 91.00 | 92.24 | 89.53 | 90.87 | 48.53 |
| | | LLaVA | 50.16 | 50.08 | **99.53** | 66.64 | 99.37 |
| | | MiniGPT-4 | 74.47 | 78.63 | 67.20 | 72.47 | 42.73 |
| | | InstructBLIP | **85.77** | **79.21** | 97.00 | **87.20** | 61.23 |
| | *Popular* | BLIP | 88.40 | 87.55 | 89.53 | 88.53 | 51.13 |
| | | LLaVA | 50.03 | 50.02 | **99.67** | 66.61 | 99.63 |
| | | MiniGPT-4 | 69.93 | **70.40** | 68.80 | 69.59 | 48.87 |
| | | InstructBLIP | **75.03** | 67.39 | 97.00 | **79.53** | 71.97 |
| | *Adversarial* | BLIP | 82.37 | 78.31 | 89.53 | 83.55 | 57.17 |
| | | LLaVA | 50.13 | 50.07 | **99.67** | 66.65 | 99.53 |
| | | MiniGPT-4 | 64.33 | **63.62** | 66.93 | 65.24 | 52.60 |
| | | InstructBLIP | **65.46** | 59.48 | 97.00 | **73.75** | 81.53 |
| GQA | *Random* | BLIP | 89.93 | 91.08 | 88.53 | 89.79 | 48.60 |
| | | LLaVA | 50.17 | 50.08 | **99.20** | 66.56 | 99.03 |
| | | MiniGPT-4 | 71.33 | **78.67** | 58.53 | 67.13 | 37.20 |
| | | InstructBLIP | **83.90** | 78.39 | 93.60 | **85.32** | 59.70 |
| | *Popular* | BLIP | 86.63 | 85.34 | 88.47 | 86.87 | 51.83 |
| | | LLaVA | 50.03 | 50.02 | **99.47** | 66.56 | 99.43 |
| | | MiniGPT-4 | 68.26 | **72.99** | 58.00 | 64.64 | 39.73 |
| | | InstructBLIP | **71.87** | 65.24 | 93.60 | **76.89** | 71.73 |
| | *Adversarial* | BLIP | 82.40 | 78.89 | 88.47 | 83.41 | 56.07 |
| | | LLaVA | 49.77 | 49.88 | **99.20** | 66.38 | 99.43 |
| | | MiniGPT-4 | 64.23 | **66.29** | 57.93 | 61.83 | 43.70 |
| | | InstructBLIP | **64.30** | 59.02 | 93.60 | **72.39** | 79.30 |

Table 11: SEEM-based POPE results of LVLMs and BLIP on A-OKVQA and GQA. The probing objects are selected from the segmentation results of SEEM on these datasets. The best results in each block except BLIP are denoted in bold. We employ the BLIP fine-tuned on VQAv2 from LAVIS (Li et al., 2022a).

| Model | POPE | BLEU-1 | BLEU-2 | METEOR | ROUGE-L |
|---|---|---|---|---|---|
| InstructBLIP | 89.29 | 59.5 | 45.2 | 22.6 | 42.3 |
| LLaVA | 68.65 | 22.0 | 13.9 | 19.8 | 22.4 |
| MiniGPT-4 | 78.86 | 41.1 | 28.8 | 25.6 | 44.7 |

Table 12: MSCOCO caption results.