# OpenReview forum: "Evaluating Object Hallucination in Large Vision-Language Models"
_EMNLP/2023/Conference — EMNLP 2023 Main_

### Official Review · Reviewer_gyaq · 2023-07-27

**Soundness:** 4

**Excitement:**

4: Strong: This paper deepens the understanding of some phenomenon or lowers the barriers to an existing research direction.

**Missing References:**



**Paper Topic And Main Contributions:**

This paper studies the object hallucination problem in large vision-language (VL) models. It first utilize CHAIR to evaluate modern LVLMs and discuss the disadvantages of CHAIR. Then, this paper proposes a new evaluation framework, named Polling-based Object Probing Evaluation (POPE). Results indicate that POPE is more stable and scalable, which could serve as a better evaluation tool for the object hallucination issue of LVLMs.

**Questions For The Authors:**

Please check "Reasons To Reject", where my questions are included.

**Reasons To Accept:**

1. The paper is well written and structured.
2. The proposed pipepline could serve as a better evaluation method than CHAIR, to evaluate LVLMs for the object hallucination issue.
3. The experiments are solid and exhaustive.


**Reasons To Reject:**

I don't see explicit reasons to reject this paper. However, some parts could be made more clear or improved:

* Where do you get the list of non-existent objects for negative sampling? Are they also from the predefined object list as used in CHAIR computation or a larger pool?
* Evaluate models with more diverse prompts and answer candidates. For example, utilizing synonyms of "yes" and "no" might help to improve evaluation accuracy.

**Reproducibility:**

3: Could reproduce the results with some difficulty. The settings of parameters are underspecified or subjectively determined; the training/evaluation data are not widely available.

**Reviewer Confidence:**

4: Quite sure. I tried to check the important points carefully. It's unlikely, though conceivable, that I missed something that should affect my ratings.

---

> ### Author Rebuttal · Authors · 2023-08-29
>
> Thank you for the careful reading and helpful feedback. We respond below to your questions and concerns:
>
> > Where do you get the list of non-existent objects for negative sampling? Are they also from the predefined object list as used in CHAIR computation or a larger pool?
>
> We utilize the same set of 80 COCO objects as used in CHAIR computation. Our analysis in Section 4 reveals that LVLMs have a tendency to hallucinate objects that are commonly seen and co-occurring. Therefore, we propose three different negative sampling strategies, which focus on random, frequently appearing, and frequently co-occurring objects respectively.
>
>
>
> > Evaluate models with more diverse prompts and answer candidates. For example, utilizing synonyms of "yes" and "no" might help to improve evaluation accuracy.
>
> Thank you for your advice. We count the number of answers in POPE_random that start with "yes" or "no". The results are presented below.
>
> | InstructBLIP | MiniGPT-4 | LLaVA     | Multimodal-GPT | mPLUG-Owl |
> | :----------- | :-------- | :-------- | :------------- | :-------- |
> | 3000/3000    | 2974/3000 | 3000/3000 | 3000/3000      | 2991/3000 |
>
> The findings indicate that a majority of answers start with either "yes" or "no", making them identifiable through a straightforward matching strategy. In our approach, we classify answers that incorporate synonyms of "no",  such as "not" and "cannot" as negative responses, while treating the rest as positive. Therefore, we believe the accuracy of our current methods is acceptable and leave further improvement such as using large language model as assistance to future work.

---

### Official Review · Reviewer_Dmv1 · 2023-07-31

**Soundness:** 4

**Excitement:**

4: Strong: This paper deepens the understanding of some phenomenon or lowers the barriers to an existing research direction.

**Justification For Ethical Concerns:**

-

**Missing References:**

- Line 53: the reference to the image captioning task is quite recent, in my opinion, for the image captioning task, it would be better to also refer to Im2text (Ordonez et. al., 2011), FLICKR 8k (Hodosh et. al., 2013), FLICKR-30k (Young et. al., 2014), MSCOCO (Lin et. al., 2014), Pinterest40M (Mao et. al., 2016), and Conceptual Captions (Sharma et. al., 2018)
- Line 54: the reference to the vqa task is quite recent, in my opinion, for the VQA task, it would be better to also refer to "VQA: Visual Question Answering" (Antol et. al., 2015) and "Yin and Yang: Balancing and Answering Binary Visual Questions" (Zhang et. al., 2016)

**Paper Topic And Main Contributions:**

The paper presents a systematic study on object hallucination of large vision language models (LVLMs), providing both qualitative and quantitative analyses to better understand object hallucination in various LVLMs.  The paper showcases the limitation of the existing object hallucination metric, CHAIR, which is unstable and requires complex parsing rules to be used. To this end, the paper proposes a new object evaluation metric, POPE, which evaluates object hallucination through a binary classification task. The paper shows the effectiveness of POPE, demonstrating better stability and scalability to various datasets. The paper also displays that object hallucination is not always aligned with the task-based evaluation performance on the VQA task, showing the importance of considering both evaluation aspects for real-world applications.

**Questions For The Authors:**

- On Figure 2a, given some objects (cat and horse) are rarely hallucinated despite having high frequency, have authors evaluated the trend of the object hallucination for the other objects (let say top-20 or top-30)? Does the trend hold in general?
- On section 4.2, do author also evaluate the co-occuring on different objects other than dining table? Does it produce a consistent trend for different objects?
- On section 4.3, what is the probing object used in HR_C@10?
- On section 4.3, since the top-10 objects only cover 50% of all the hallucinated objects, what about the remaining objects? Does it produce a downward trend depending on the frequency of the objects in MSCOCO?
- On section 5.4, why don't author compare POPE with other VL tasks such as image captioning or image classification? Is there any specific reason why authors choose VQA? It will be better if authors can also compare with performance on other tasks before making the conclusion on Line 512.

**Reasons To Accept:**

- The paper clearly explained the problem with the existing object hallucination metric, CHAIR, which is unstable over different prompt and requires complex human-crafted parsing rules to be used
- The paper proposes a new metric, POPE, which solves the instability and complexity problems of CHAIR, by framing object hallucination into as a binary classification task. The stability and scalability of POPE is clearly evaluated through a series of experiment over multiple datasets.

**Reasons To Reject:**

- The qualitative and quantitative analysis (Section 4.2 and 4.3) only shows the top-10 objects in MSCOCO, it is not enough to support the two hypotheses, it would be better to show the cumulative hallucinated count over all objects in MSCOCO for the first hypothesis, especially when there are some objects in the top-10 list (cat, horse, and toilet) while for the second hypothesis, taking only a single object (e.g., "dining table") for analysis is inadequate to provide a robust supporting evidence to conclude the hypothesis.

**Reproducibility:**

4: Could mostly reproduce the results, but there may be some variation because of sample variance or minor variations in their interpretation of the protocol or method.

**Reviewer Confidence:**

4: Quite sure. I tried to check the important points carefully. It's unlikely, though conceivable, that I missed something that should affect my ratings.

**Typos Grammar Style And Presentation Improvements:**

- For section 4.3, it would be better if authors can show the cumulative hallucination count of all objects in MSCOCO sorted by frequency
- For section 5.4, it will be better if authors can also compare with performance on tasks other than VQA

---

> ### Author Rebuttal · Authors · 2023-08-29
>
> Thank you for the careful reading and helpful feedback. We respond below to your questions and concerns:
>
> > On Figure 2a, given some objects (cat and horse) are rarely hallucinated despite having high frequency, have authors evaluated the trend of the object hallucination for the other objects (let say top-20 or top-30)? Does the trend hold in general?
> >
> > For section 4.3, it would be better if authors can show the cumulative hallucination count of all objects in MSCOCO sorted by frequency
>
> Yes, the same trend holds for other objects as well. Due to space limitations, we have only listed the trend for the top 10 objects in our figure. Here, we present the complete statistics for all 80 COCO objects.. We organize these objects based on their appearance frequency in the COCO validation set and calculate their hallucination times in groups of 10. The results are displayed in the table below.
>
> |                | Top 1-10 | Top 11-20 | Top 21-30 | Top 31-40 | Top 41-50 | Top 51-60 | Top 61-70 | Top 71-80 |  Sum |
> | -------------- | -------: | --------: | --------: | --------: | --------: | --------: | --------: | --------: | ---: |
> | LLaVA          |     1349 |       329 |       263 |        85 |       241 |       245 |       179 |        92 | 2783 |
> | MiniGPT-4      |      598 |       136 |       206 |        99 |        83 |       118 |        58 |        32 | 1330 |
> | Multimodal-GPT |      690 |       209 |       204 |        63 |       141 |       136 |       112 |        47 | 1602 |
> | mPLUG-Owl      |     2860 |       546 |       482 |       130 |       527 |       247 |       172 |        63 | 5027 |
>
> We also list the cumulative proportions of the hallucinated objects below.
>
> |                | Top 10 | Top 20 | Top 30 | Top 40 | Top 50 | Top 60 | Top 70 | Top 80 |
> | -------------- | -----: | -----: | -----: | -----: | -----: | -----: | -----: | -----: |
> | LLaVA          |  48.47 |  60.29 |  69.74 |  72.80 |  81.46 |  90.26 |  96.69 |    100 |
> | MiniGPT-4      |  44.96 |  55.19 |  70.68 |  78.12 |  84.36 |  93.23 |  97.59 |    100 |
> | Multimodal-GPT |  43.07 |  56.12 |  68.85 |  72.78 |  81.59 |  90.07 |  97.07 |    100 |
> | mPLUG-Owl      |  56.89 |  67.75 |  77.34 |  79.93 |  90.41 |  95.33 |  98.75 |    100 |
>
> From the table, we can see that the hallucination times decrease across different groups (except for top 31-40), and the top-10 objects consistently exhibit the highest hallucination times.  Additionally, across all models,  the top-30 objects occupies approximately 70% of the hallucinated objects. Therefore, it is reasonable to conclude that LVLMs tends to hallucinate frequently appearing objects in the instruction corpus.
>
>
>
> > On section 4.2, do author also evaluate the co-occurring on different objects other than dining table? Does it produce a consistent trend for different objects?
>
> Yes, the trend hold for other objects too. Due to space limitations, we only list the result of dining table. Here, we provide the full statistics of dining table, chair and car.
>
> | dining table   | Top 1-10 | Top 11-20 | Top 21-30 | Top 31-40 | Top 41-50 | Top 51-60 |  >60 |  Sum |
> | -------------- | -------: | --------: | --------: | --------: | --------: | --------: | ---: | ---: |
> | LLaVA          |       88 |        19 |        17 |        11 |         6 |         4 |    0 |  145 |
> | MiniGPT-4      |       32 |         8 |         6 |         1 |         2 |         2 |    0 |   50 |
> | Multimodal-GPT |       84 |        34 |        17 |        17 |         2 |         3 |    0 |  157 |
> | mPLUG-Owl      |      234 |        36 |        13 |        20 |        20 |         3 |    0 |  326 |
>
> | chair          | Top 1-10 | Top 11-20 | Top 21-30 | Top 31-40 | Top 41-50 | Top 51-60 |  >60 |  Sum |
> | -------------- | -------: | --------: | --------: | --------: | --------: | --------: | ---: | ---: |
> | LLaVA          |       90 |        30 |        20 |        25 |         7 |         3 |    7 |  182 |
> | MiniGPT-4      |       12 |         4 |         0 |         2 |         1 |         0 |    0 |   19 |
> | Multimodal-GPT |       48 |        28 |        11 |         6 |         2 |         0 |    3 |   98 |
> | mPLUG-Owl      |      427 |        65 |        83 |       117 |        11 |         3 |    9 |  715 |
>
> | car            | Top 1-10 | Top 11-20 | Top 21-30 | Top 31-40 | Top 41-50 | Top 51-60 |  >60 |  Sum |
> | -------------- | -------: | --------: | --------: | --------: | --------: | --------: | ---: | ---: |
> | LLaVA          |       90 |        17 |        10 |         7 |         1 |         0 |    2 |  127 |
> | MiniGPT-4      |       18 |         1 |         2 |         1 |         0 |         0 |    1 |   23 |
> | Multimodal-GPT |       19 |         4 |         0 |         0 |         1 |         0 |    0 |   24 |
> | mPLUG-Owl      |      201 |        44 |        20 |         1 |         5 |         2 |    2 |  275 |
>
> From the tables above, we can observe that the descending trends remain consistent for all three objects.Therefore, it is reasonable to conclude that LVLMs tends to hallucinate frequently co-occurring objects in the instruction corpus.
>
>
>
> > On section 4.3, what is the probing object used in HR_C@10?
>
> The probing objects for HR_C@10 is dining table. We will rectify this in the following revisions.
>
>
>
> > On section 4.3, since the top-10 objects only cover 50% of all the hallucinated objects, what about the remaining objects? Does it produce a downward trend depending on the frequency of the objects in MSCOCO?
>
> Yes, this downward trend also applies to less frequent objects. To confirm this, we add supplementary experiments for HR_A and HR_C with the top 20 and top 30 objects. Additionally, for HR_C, we've included the results for two more probing objects (*i.e.,* chair and car). The results are displayed in the tables below. The data in parentheses indicate the difference in HR compared to the preceding column, corresponding to the HR within current interval (*e.g.,* top 11-20 or top 21-30).
>
> |                | HR_A@10        | HR_A@20        | HR_A@30        |
> | -------------- | -------------- | -------------- | -------------- |
> | mPLUG-Owl      | 0.5455(0.5455) | 0.6591(0.1136) | 0.7533(0.0942) |
> | LLaVA          | 0.4620(0.4620) | 0.5911(0.1291) | 0.6796(0.0855) |
> | MultiModal-GPT | 0.4152(0.4152) | 0.5399(0.1247) | 0.6743(0.1344) |
> | MiniGPT-4      | 0.4610(0.4610) | 0.5758(0.1148) | 0.7207(0.1449) |
>
> | dining table   | HR_C@10        | HR_C@20        | HR_C@30        |
> | -------------- | -------------- | -------------- | -------------- |
> | mPLUG-Owl      | 0.6608(0.6608) | 0.7926(0.1318) | 0.8253(0.0597) |
> | LLaVA          | 0.5628(0.5628) | 0.7329(0.1701) | 0.8595(0.1266) |
> | MultiModal-GPT | 0.5742(0.5742) | 0.7849(0.2107) | 0.8961(0.1112) |
> | MiniGPT-4      | 0.5600(0.5600) | 0.6980(0.1380) | 0.9145(0.2165) |
>
> | chair          | HR_C@10        | HR_C@20        | HR_C@30        |
> | -------------- | -------------- | -------------- | -------------- |
> | mPLUG-Owl      | 0.5926(0.5926) | 0.7186(0.1260) | 0.8201(0.1015) |
> | LLaVA          | 0.5206(0.5206) | 0.6830(0.1624) | 0.8152(0.1322) |
> | MultiModal-GPT | 0.5732(0.5732) | 0.7576(0.1844) | 0.8811(0.1235) |
> | MiniGPT-4      | 0.5701(0.5701) | 0.7746(0.2045) | 0.8581(0.0835) |
>
> | car            | HR_C@10        | HR_C@20        | HR_C@30        |
> | -------------- | -------------- | -------------- | -------------- |
> | mPLUG-Owl      | 0.7587(0.7587) | 0.9136(0.1549) | 0.9707(0.0571) |
> | LLaVA          | 0.6870(0.6870) | 0.8886(0.2016) | 0.9188(0.0302) |
> | MultiModal-GPT | 0.6031(0.6031) | 0.8482(0.2451) | 0.8623(0.0141) |
> | MiniGPT-4      | 0.6444(0.6444) | 0.8278(0.1834) | 0.9417(0.1139) |
>
> The above results show that HR_A and HR_C in most interval keep descending along with the co-occurring frequency of objects, and all of them are siginificantly smaller than HR_A/HR_C@10. Hence, it's reasonable to conclude that LVLMs are more susceptible to hallucinating objects that appear frequently or co-occur in the instruction corpus.
>
>
>
> > On section 5.4, why don't author compare POPE with other VL tasks such as image captioning or image classification? Is there any specific reason why authors choose VQA? It will be better if authors can also compare with performance on other tasks before making the conclusion on Line 512.
>
> We select VQA tasks as they offer a comprehensive assessment of the advanced capabilities of LVLMs. VQA tasks involve posing questions about images, demanding a profound understanding of both visual and language content. Compared to tasks like image classification and image captioning, challenging VQA tasks such as A-OKVQA and GQA can better showcase the formidable image comprehension and reasoning capabilities of LVLMs. Following GPT-4 [1], recent LVLMs [2, 3] also demonstrate their abilities by providing case studies on VQA-like examples, such as explaining jokes or story generation given an image.
>
> Following your advice, we also add experiments on image captioning tasks. Specifically, we ask InstructBLIP, MiniGPT-4, and LLaVA to generate captions for 500 images utilized in POPE. Subsequently, we evaluated these captions using BLEU-1, BLEU-2, METEOR, and ROUGE-L. The evaluation results are presented below:
>
> |              | BLEU-1 | BLEU-2 | METEOR | ROUGE-L |
> | ------------ | -----: | -----: | -----: | ------: |
> | InstructBLIP |   59.5 |   45.2 |   22.6 |    42.3 |
> | MiniGPT-4    |   41.1 |   28.8 |   25.6 |    44.7 |
> | LLaVA        |   22.0 |   13.9 |   19.8 |    22.4 |
>
>  The results show that their performance on image captioning are basically consistent with POPE.
>
>
>
> > Missing References
>
> Thank you for your suggestions. We will include these reference in the subsequent versions.
>
>
>
> [1] OpenAI. GPT-4 Technical Report. arXiv preprint arXiv: 2303.08774, 2023.
>
> [2] Ye Q, et al. mplug-owl: Modularization empowers large language models with multimodality[J]. arXiv preprint arXiv:2304.14178, 2023.
>
> [3] Zhu D, et al. Minigpt-4: Enhancing vision-language understanding with advanced large language models[J]. arXiv preprint arXiv:2304.10592, 2023.

---

### Official Review · Reviewer_tq5W · 2023-08-09

**Soundness:** 3

**Excitement:**

3: Ambivalent: It has merits (e.g., it reports state-of-the-art results, the idea is nice), but there are key weaknesses (e.g., it describes incremental work), and it can significantly benefit from another round of revision. However, I won't object to accepting it if my co-reviewers champion it.

**Paper Topic And Main Contributions:**

The work presents the first systematic study on object hallucination of LVLMs. The authors conduct the evaluation experiments on several representative LVLMs. They also design a polling-based query method called POPE for better evaluation of object hallucination.

**Questions For The Authors:**

1. The large model is unlikely to only reply with yes/no. How to handle subsequent replies and whether there will be inconsistencies between the previous and the current responses remains to be seen.
2. Is Table 6 based on a subset or the entire VQA dataset, and does the evaluation for VQA task also use the Yes/No scoring method?

**Reasons To Accept:**

1. The paper is well written and easy to follow.
2. The work conducts an empirical study on object hallucination for several representative LVLMs, and are highly affected by object hallucination.
3. The work proposes an object hallucination evaluation method, which can be easily extended to unannotated datasets.

**Reasons To Reject:**

1. Is object hallucination the most important problem of multimodal LLM? Others include knowledge, object spatial relationships, fine-grained attributes, etc.
2. Is it sufficient to measure object hallucination through only yes/no responses? Yes response does not necessarily indicate that the model comprehends the presence of the object in the image, as it may still produce incorrect objects when undertaking other tasks.

**Reproducibility:**

4: Could mostly reproduce the results, but there may be some variation because of sample variance or minor variations in their interpretation of the protocol or method.

**Reviewer Confidence:**

5: Positive that my evaluation is correct. I read the paper very carefully and I am very familiar with related work.

---

> ### Author Rebuttal · Authors · 2023-08-29
>
> Thank you for the careful reading and helpful feedback. We respond below to your questions and concerns:
>
> > Is object hallucination the most important problem of multimodal LLM? Others include knowledge, object spatial relationships, fine-grained attributes, etc.
>
> Thanks for your insightful comment. Actually, object hallucination is one of the most fundamental issues that degrade the performance of LVLMs. It is because the semantic understanding of an image is often directly related to objects that it contains, those (object spatial relationship, fine-grained attributes) as you mentioned. Furthermore, a number of visual-text tasks are also defined over objects, *e.g.,* VQA and visual reasoning. Evaluating object hallucination essentially reflects whether the model can accurately identify the main objects in an image, which is the basic ability of LVLMs to understand images, providing the foundation for addressing complex tasks. However, as the reviewer suggested, a future direction is to extend the study of object hallucination to a more fine-grained hallucination (e.g., attribute-level hallucination).
>
> Following the suggestion of the reviewer, we conduct statistical analyses on finer-grained hallucination issues of LVLMs. Concretely, we select 10 captions from each of the 5 LVLMs and manually annotate their hallucination frequency on object, knowledge, spatial ralationships and fine-grained attributes. The results are presented below.
>
> |                | object | knowledge | spatial ralationships | fine-grained attributes |
> | -------------- | -----: | --------: | --------------------: | ----------------------: |
> | mPLUG-Owl      |     10 |         1 |                     2 |                       7 |
> | LLaVA          |      8 |         0 |                     3 |                       3 |
> | MultiModal-GPT |      6 |         0 |                     0 |                       6 |
> | MiniGPT-4      |      6 |         0 |                     0 |                       3 |
> | Sum            |     30 |         1 |                     5 |                      19 |
>
> The above statistics show that object hallucination is the most prevalent one, while knowledge and spatial hallucination only occupy a small part. We also notice that LVLMs often make mistakes on fine-grained attributes of objects, such as number, color or textual content. However, it is challenging to enumerate all fine-grained attributes, and their annotation and evaluation are also hard to be efficiently performed. As a comparison, the object halluciation issue is much easier to be evaluated, and their annotations can be acquired from the caption data or through a segmentation model (*e.g.,* SEED).
>
>
>
> > Is it sufficient to measure object hallucination through only yes/no responses? Yes response does not necessarily indicate that the model comprehends the presence of the object in the image, as it may still produce incorrect objects when undertaking other tasks.
>
> We believe that the yes/no responses sufficiently reflect LVLMs' understanding of the presence of objects in the image. To verify this, we design two extra experiments to measure the consistency between the yes/no responses and captions. We find that objects which get "No" responses from models are hardly appear in models' captions, and models prefer to answer "Yes" to objects mentioned in captions.
>
> 1. Consistency between "No" responses and captions
>
>    We first collect objects and corresponding images from POPE_random which get a "No" answer from models. Then, we ask models to generate captions on these images and count the occurence of these objects. We collect data from InstructBLIP and MiniGPT-4, since their yes/no proportions are relatively more balanced. The results are presented in the table below.
>
>    | Model        | Number of No responses | Number of objects in captions | Ratio |
>    | :----------- | ---------------------: | ----------------------------: | ----: |
>    | InstructBLIP |                   1303 |                             0 | 0.00% |
>    | MiniGPT-4    |                   1445 |                             5 | 0.35% |
>
> 2. Consistency between "Yes" responses and captions
>
>    We first collect 500 captions from each model and extract COCO objects from them. Then, we form them into POPE questions using the template "Is there a \<object\> in the image?". We ask the models about these questions and count the number of "Yes" responses. For the same reason, we only test InstructBLIP and MiniGPT-4. The results are presented in the table below.
>
>    | Model        | Number of objects in captions | Number of Yes responses | Ratio |
>    | :----------- | ----------------------------: | ----------------------: | ----: |
>    | InstructBLIP |                           664 |                     664 |  100% |
>    | MiniGPT-4    |                          1034 |                     961 | 92.9% |
>
> The above experiments show that the yes/no responses and captions of LVLMs are highly consistent. It also indicates the effectiveness of the yes/no responses in POPE to evaluate the object hallucination of LVLMs as existing methods.
>
>
>
>
>
> > The large model is unlikely to only reply with yes/no. How to handle subsequent replies and whether there will be inconsistencies between the previous and the current responses remains to be seen.
>
> We evaluate the consistency between the yes/no and susequent replies of models with the assistance of ChatGPT. The results show that most answers of LVLMs are consistent.
>
> Concretely, we select 200 answers of POPE questions from each model and ask ChatGPT to judge whether the two part of the answer have consistent semantic. The prompt we provide to ChatGPT is as follows:
>
> - *I will provide you with a dialogue consisting of a question and an answer. The question will be a simple closed-ended question, such as "Is there a truck in the image?" The answer will be composed of two parts: the first part will be a single 'Yes' or 'No,' and the second part will provide additional explanation or description for the reasons behind the given answer. Your task is to determine whether the attitude of both parts of the answer is consistent.*
>
>   *Here are some examples to illustrate the task:*
>
>   *Question: Is there a car in the image? Answer: Yes, there is a car in the image. Judgment: Consistent*
>
>   *Question: Is there a boat in the image? Answer: Yes, there is a car in the image. Judgment: Inconsistent*
>
>   *Question: Is there a boat in the image? Answer: No, there is a boat in the image. Judgment: Inconsistent*
>
>   *Please provide your judgment for the following dialogue:*
>
> To avoid misjudgement, we also manually check the result. The final results are presented below.
>
> | Model            | InstructBLIP |   LLaVA | MiniGPT-4 | Mulitmodal-GPT | mPLUG-Owl |
> | ---------------- | -----------: | ------: | --------: | -------------: | --------: |
> | Consistent ratio |      200/200 | 189/200 |   199/200 |        199/200 |   199/200 |
>
> The results show that most reply of LVLMs are consistent, so we believe it is reasonable to judge model's attitude with the yes/no part.
>
>
>
> > Is Table 6 based on a subset or the entire VQA dataset, and does the evaluation for VQA task also use the Yes/No scoring method?
>
> 1. We use the entire A-OKVQA and a subset of GQA (500 examples).
>
> 2. No, we use the VQA score [1] to evaluate VQA tasks with the assistance of ChatGPT. Due to page limits, we have left this part in Appendix C.
>
>    Since the reply of LVLMs are usually open-ended,  conventional evaluation for VQA tasks may not effectively measure the quality of answers. Therefore, we use ChatGPT to judge whether the model's answer matches candidate answers. The prompt we provide to ChatGPT is as follows:
>
>    *You are an examiner who can judge whether a student's answer matches the correct answers. Next, I will provide you with 10 correct answers in the form of a list and a student's answer. Please judge whether the student's answer matches one of the 10 correct answers. If it matches, please output the correct answer directly (must be an element in the list, if it matches multiple correct answers, please output the most frequent occurrence in the list); if not, please output \<NAN\> directly. Do NOT output anything else!*
>
>    *correct answers:*
>
>    *student answer:*
>
>    Then, we calculate the VQA score using the original equation $\text{score}(ans) = \min(\frac{\text{humans that said} \ ans}{3}, 1)$.
>
> ​
>
> [1] Antol S, et al. Vqa: Visual question answering[C]//Proceedings of the IEEE international conference on computer vision. 2015: 2425-2433.

---

### Meta-Review · Area_Chair_GDv6 · 2023-09-15

**Recommendation:** 5

**Metareview:**

Three reviewers provided feedback for this work and were in consensus. They found the paper to be well written, the empirical study to be well done and well presented and the new metric to be sound. There were very few complaints for this paper. One complaint was questioning if object hallucinations were an important problem in todays VLMs. A second complaint was about using yes/no responses to measure this problem. And a third complaint was about expanding to all COCO categories. The rebuttals provided by the authors were very thorough and in my opinion have addressed all these concerns.

---

### Decision · Program_Chairs · 2023-10-07

**Decision:**

Accept-Main

**Comment:**

Three reviewers provided feedback for this work and were in consensus. They found the paper to be well written, the empirical study to be well done and well presented and the new metric to be sound. There were very few complaints for this paper. One complaint was questioning if object hallucinations were an important problem in todays VLMs. A second complaint was about using yes/no responses to measure this problem. And a third complaint was about expanding to all COCO categories. The rebuttals provided by the authors were very thorough and in my opinion have addressed all these concerns.